# Transient eco-evolutionary dynamics early in a phage epidemic have strong and lasting impact on the long-term evolution of bacterial defences

Bridget Nora Janice Watson[1]*, Elizabeth Pursey[1], Sylvain Gandon[2], Edze Rients Westra[1]

1 ESI, Biosciences, University of Exeter, Cornwall Campus, Penryn, United Kingdom, 2 Centre d'Ecologie Fonctionnelle et Evolutive (CEFE), UMR 5175, CNRS-Université de Montpellier-Université Paul-Valéry Montpellier-EPHE, Montpellier, France

☉ These authors contributed equally to this work.
* b.watson3@exeter.ac.uk

**Data Availability Statement:** Data and code are available online https://doi.org/10.5281/zenodo.8193506.

## Abstract

Organisms have evolved a range of constitutive (always active) and inducible (elicited by parasites) defence mechanisms, but we have limited understanding of what drives the evolution of these orthogonal defence strategies. Bacteria and their phages offer a tractable system to study this: Bacteria can acquire constitutive resistance by mutation of the phage receptor (surface mutation, *sm*) or induced resistance through their CRISPR-Cas adaptive immune system. Using a combination of theory and experiments, we demonstrate that the mechanism that establishes first has a strong advantage because it weakens selection for the alternative resistance mechanism. As a consequence, ecological factors that alter the relative frequencies at which the different resistances are acquired have a strong and lasting impact: High growth conditions promote the evolution of *sm* resistance by increasing the influx of receptor mutation events during the early stages of the epidemic, whereas a high infection risk during this stage of the epidemic promotes the evolution of CRISPR immunity, since it fuels the (infection-dependent) acquisition of CRISPR immunity. This work highlights the strong and lasting impact of the transient evolutionary dynamics during the early stages of an epidemic on the long-term evolution of constitutive and induced defences, which may be leveraged to manipulate phage resistance evolution in clinical and applied settings.

## Introduction

Organisms have evolved a large repertoire of defence systems that offer protection against infectious diseases. Some of these defences are always active—known as constitutive defences—whereas others are elicited by parasites—known as inducible defences [1]. Fitness trade-offs associated with these defences tend to manifest accordingly (i.e., constitutive or infection-induced [2,3]), and, consequently, organisms are predicted to invest more in constitutive defences as the infection risk increases, and less in induced defences [4]. Yet, it is not clear

**Funding:** This work was funded by a grant from the Natural Environment Research Council (NE/S001921/1)(http://gotw.nerc.ac.uk/list_full.asp?pcode=NE%2FS001921%2F1&cookieConsent=A) and a grant from the European Research Council (https://erc.europa.eu) (ERC-STG-2016-714478 - EVOIMMECH) awarded to E.R.W. B.N.J.W acknowledges support from the Biotechnology and Biological Sciences Research Council (BB/X010600/1)(https://www.ukri.org/councils/bbsrc/) The funders had no role in study design, data collection and analysis, decision to publish, or preparation of the manuscript.

**Competing interests:** The authors have declared that no competing interests exist.

**Abbreviations:** CFU, colony-forming unit; dpi, day postinfection; PFU, plaque-forming unit; sm, surface mutation.

what influences the initial evolution of each resistance strategy, or if there are interactions between these alternative strategies that influence their long-term coevolution.

Bacteria and their viruses, called bacteriophages or phages, are a useful model system to study the evolution of different defence strategies. Bacteria can evolve resistance against phages through a wide range of mechanistically distinct defence strategies [5]. Many of those provide innate immunity and are therefore key for determining the levels of preexisting phage resistance but are less important for the evolution of de novo phage resistance [6]. Rapid evolution of phage resistance typically relies on either mutation of the phage receptor, in order to prevent phage adsorption to the bacterial cell, or the acquisition of CRISPR-Cas adaptive immunity, which is based on insertion of phage-derived sequences (spacers) into CRISPR loci in the host genome that are used as a genetic memory to detect and destroy phages during reinfection [7]. Evolution of phage resistance by the opportunistic human pathogen *Pseudomonas aeruginosa* PA14 against its obligatory lytic phage, DMS3*vir*, is a case in point, relying either on mutation of the Type IV pilus (surface mutation, *sm*), or its type I-F CRISPR-Cas system. Mutations in the Type IV pilus carry a constitutive cost of resistance, and although CRISPR-Cas systems can be costly to express and maintain in some hosts [8], in *P. aeruginosa*, CRISPR immunity is only associated with an infection-induced cost [4,9,10]. Since both mutations confer (almost) perfect resistance to phages, cells that carry these 2 mutations would not benefit from higher fitness because resistance would not be much higher. In other words, there is strong negative epistasis in fitness between these mutations. This negative epistasis is known to strongly influence the trajectories of adaptation [11–13]. In accordance with theory, selection favours CRISPR immune bacteria over surface mutants when phage densities are low, but the balance tips in favour of surface mutants as phage densities increase [4]. In this study, we combine theory and experiments with this bacteria–phage model to explore if and how the short-term transient evolutionary dynamics of these different resistances impacts their long-term evolution.

## Results

To address this gap in our knowledge, we first generated a mathematical model to identify the key parameters that influence the transient evolution of defence mechanisms in an initially sensitive bacterial population when it is exposed to phages (**Fig 1**). The model allows for the joint evolution of the 2 mechanisms of resistance that can be acquired by susceptible cells via mutation (surface mutation resistance, *sm*) or acquisition of a new spacer (CRISPR resistance) and accounts for possible costs of resistance (fixed cost for surface resistance and conditional cost for CRISPR resistance). This deterministic model allows us to track how initial conditions affect the epidemiology and evolution of the system and thus make predictions on the final frequency of the different types of resistance (more details of the model are presented in **S1 Text**). Analysing the change in frequency of the different resistance forms revealed that the initial phase of evolution is key. The more rapidly CRISPR immune bacteria increase in frequency, the more they interfere with selection for surface resistance for 2 main reasons. First, the increase in resistance to phages in the bacterial population reduces the fitness benefit associated with a new resistance mechanism. Second, the increase in resistance to phages feeds back on viral dynamics, and the drop in phage density reduces the selective pressure for *sm*. This causes the spread of this alternative form of resistance to slow down (interference in the selection coefficients is derived in the **S1 Text**). Consequently, factors that increase the early acquisition of spacers (relative to the acquisition of mutations in phage receptor genes) will promote the evolution of CRISPR-based immunity and vice versa for the evolution of surface-based resistance (**Figs 2 and S1**). Since the acquisition of receptor mutations is tightly linked

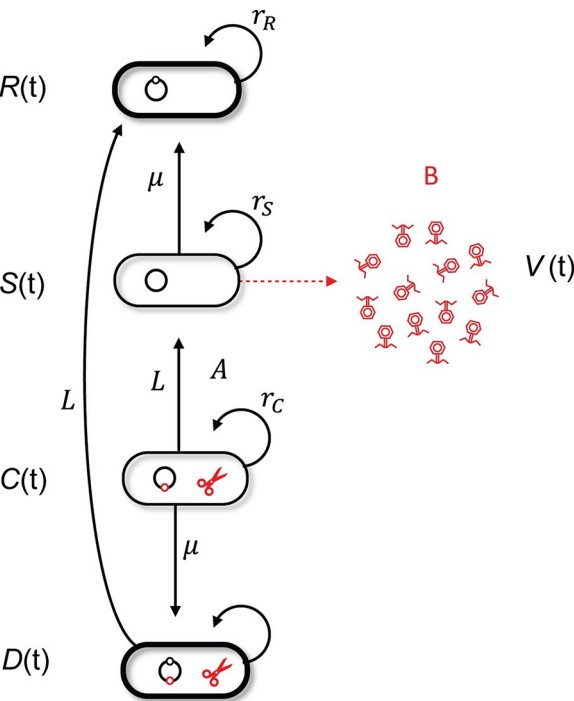

**Fig 1. Schematic representation of the model.** Naive and uninfected hosts (S hosts) reproduce at rate $r_S$. Upon infection by the phages, they release a burst size $B$ of new viral particles. Two distinct types of resistance may emerge: surface modification ($R$ hosts) or CRISPR resistance ($C$ hosts). $R$ hosts reproduce at a rate $r_R = e^{-c_R} r_S$, where $c_R$ measures the cost of resistance. $C$ hosts reproduce at a rate $r_C = e^{-\tau a V} r_S$, where $\tau$ measures the toxicity induced by CRISPR immunity when resistant cells are exposed to the virus. $D$ hosts reproduce at a rate $r_D = e^{-c_R} r_S$ because even though they have both types of resistance, they only express the cost of constitutive resistance because they are never infected by phages. Cells acquire surface modification at rate $\mu$ (this rate is constant) and acquire CRISPR resistance at rate $AaV$ (this rate varies with the exposition to viral particles).

to bacterial replication, a key prediction from the model is therefore that the amount of replication that can occur in the initially sensitive population until carrying capacity is reached has a major impact on the type of resistance that emerges (**Figs 2A-2C and S3**). Another prediction from the model is the lack of double resistance (**S1–S4 Figs**). Indeed, double resistance has the same level of resistance as single resistance, and it carries the same cost as surface resistance. This implies that epistasis between the 2 resistance mechanisms is strongly negative. This negative epistasis is expected to yield negative linkage disequilibrium (**S4 Fig**) and a low frequency of double resistance.

Next, we performed evolution experiments with *P. aeruginosa* PA14 and its nonlysogenic phage, DMS3*vir*, to test this model prediction. *P. aeruginosa* PA14 carries a type I-F CRISPR-Cas immune system [14] and is commonly used as a model to study the evolutionary ecology of CRISPR-Cas systems [4,7]. We inoculated the same volume of fresh growth media with different amounts of sensitive cells from an overnight culture of WT PA14, ranging from 10% to 0.01% of the total final volume (**Fig 2**). As a result, the number of rounds of replication until the cultures reached carrying capacity differed between treatments, which, in turn, affected the opportunity for *sm* clones to emerge in those cultures. Each bacterial culture was also infected with phage DMS3*vir* ($10^5$ plaque-forming units (PFUs) ml$^{-1}$). Despite the differences in initial cell concentrations, all cultures reached similar final counts (**Fig 2D**) as did the phage counts (**Fig 2E**). Even though the treatments with a larger inoculum were more likely to initially carry

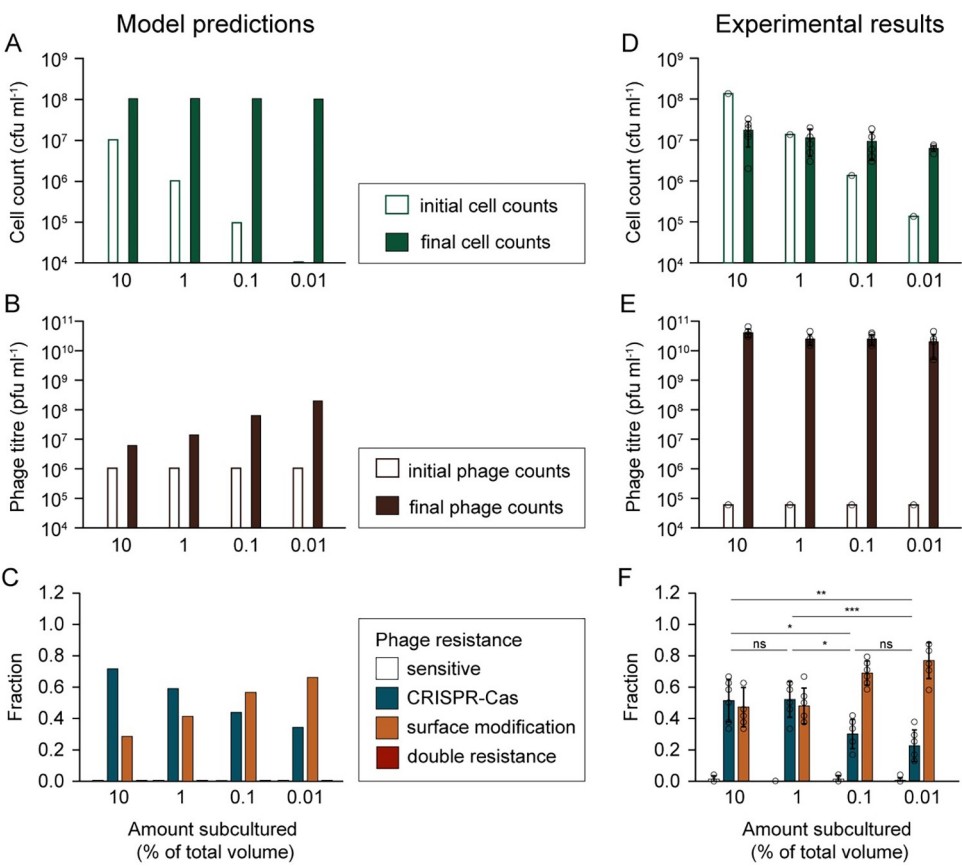

**Fig 2. The emergence of surface mutants is replication dependent.** Cultures were inoculated with different proportions of stationary phase, sensitive WT *P. aeruginosa* (600 μl: 10%, 60 μl: 1%, 6 μl: 0.1%, and 0.6 μl: 0.01% of final volume), and $10^5$ pfu ml$^{-1}$ DMS3*vir* phages. Plots show: (**A**, **D**) cell counts and (**B**, **E**) phage counts for each treatment (white bars: initial, green/ purple bars: final (1 day)), as well as (**C**, **F**) the fraction of each resistance type was determined (for 24 clones per replicate, white: phage sensitive, blue: CRISPR-Cas immune, orange: surface-based resistance). (**A-C**) show the outcomes predicted by the model (see S1 Text) and (**D-F**) show the experimental results. Data shown are the mean ± 1 standard deviation, 6 replicates per treatment. (**F**) Statistical significance between the fractions of CRISPR resistance for each treatment was testing using ANOVA with post hoc Tukey test, 10% vs. 1%: $p = 0.9995$ (ns, not significant), 10% vs. 0.1%: $p = 0.0164$ (*), 10% vs. 0.001%: $p = 0.0011$ (**), 1% vs. 0.1%: $p = 0.0130$ (*), 1% vs. 0.01%: $p = 0.0009$ (***), 0.1% vs. 0.01%: $p = 0.6463$ (ns). Data are available at https://doi.org/10.5281/zenodo.8193506.

*sm* clones (due to more standing genetic variation), as predicted by the model (**Figs 2A–2C** and **S3**), cultures with the smallest inoculum of 0.01% mostly evolved surface-based resistance (*sm* fraction: 0.77 ± 0.11), whereas those with the largest inoculum of 10% mostly evolved CRISPR immunity (CRISPR fraction: 0.51 ± 0.14) (**Fig 2F**). The proportion of CRISPR immunity that evolved significantly increased with the bacterial inoculum size (ANOVA, $p = 0.0002$, with post hoc Tukey test, 10% vs. 1%: $p = 0.9995$, 10% vs. 0.1%: $p = 0.0164$, 10% vs. 0.001%: $p = 0.0011$, 1% vs. 0.1%: $p = 0.0130$, 1% vs. 0.01%: $p = 0.0009$, 0.1% vs. 0.01%: $p = 0.6463$). These data therefore show that higher levels of CRISPR immunity emerged when less bacterial replication occurred.

Our model also predicts that increasing phage infection will promote the evolution of CRISPR-Cas immunity. This is because CRISPR immunity and the incorporation of new spacers in the CRISPR array requires phage infection, whereas the mutation process that yields surface mutants is independent of phage infection; increasing the initial phage dose thus increases

the early acquisition of CRISPR immunity relative to the acquisition of surface resistance. Likewise, when bacteria grow to higher population densities, phages will reach higher densities, more rapidly, as they will be more likely to find a host when they diffuse through the media (**S1 Fig**). Note, however, that because CRISPR immunity is associated with an infection-induced toxicity cost [4], selection favours bacteria with surface resistance when the density of viruses becomes too high (without toxicity cost, CRISPR immune bacteria are always favoured at higher viral doses; see **S2 Fig**).

To test these model predictions, we experimentally examined, using a full factorial design, the effect of phage exposure by varying the initial doses ($10^1$, $10^2$, $10^3$, $10^5$, and $10^9$ PFU ml$^{-1}$) and the carrying capacity of the media (0.0002, 0.002, 0.02, and 0.2% glucose; **S5A Fig**) in infection experiments of PA14 with DMS3*vir*. We analysed the cell and phage densities across the experiment and saw that both conditions had a significant impact on the phage densities at t = 1, 2, and 3 days postinfection (dpi), as predicted (**S5A** and S5B **Fig**). In conditions with low phage doses and/or low carrying capacity (low glucose concentration), phage counts increased over the duration of the experiment, since the phage epidemic spread slowly due to the low initial phage counts and/or the relative sparseness of bacterial hosts (**S5C Fig**). Whereas in conditions with high phage doses and/or high carrying capacities, phage densities peaked with high counts by t = 2 dpi (**S5C Fig**). Resistance profiles were then determined daily for 3 days to capture the dynamics of resistance evolution across these treatments. Visual inspection of the resulting data supported the model prediction: Generally, populations appear to evolve higher levels of CRISPR immunity when either the initial phage dose or the carrying capacity is high (**Fig 3A**). For example, following infection with $10^1$ PFU of DMS3*vir* (in high glucose conditions), CRISPR immunity was barely detected at 1 dpi (CRISPR fraction $10^1$: 0.09 ± 0.10, mean ± 1 standard deviation) but with doses of $10^3$ PFU and higher, most clones had CRISPR immunity at 1 dpi (CRISPR fraction $10^3$: 0.59 ± 0.14 $10^5$: 0.70 ± 0.10, $10^9$: 0.64 ± 0.02) (**Figs 3A and S6A**). Similarly, for the carrying capacity conditions, in the treatments with low carrying capacity (0.0002% and 0.002% glucose), CRISPR clones were not detected 1 dpi, but with high carrying capacity, CRISPR immunity was dominant (CRISPR fraction 0.02% glucose: 0.49 ± 0.25, 0.2% glucose: 0.59 ± 0.14) (**S6B Fig**). To analyse these data more rigorously, we developed statistical models.

We used separate mixed effects models to assess the relative contributions of each variable (glucose concentration, initial phage dose, final phage density, and final cell density) for each time point, controlling for treatment replicate. Model selection was then used to determine which variables were most important for explaining CRISPR evolution at each time point (**Fig 3B–3D**, see S1 and S2 Tables). At 1 dpi, final phage density and the initial phage dose were the most important variables explaining the probability that a clone evolves CRISPR, with more CRISPR evolution predicted as these variables increased (**Fig 3B**). At 2 dpi, final phage density and final cell density explained the most variation, with higher phage and cell densities both predicting an increased probability of CRISPR immunity (**Fig 3C**). Finally, by 3 dpi, initial phage dose, final phage density, and final cell density were all retained in the model (**Fig 3D**). In this case, higher phage doses at the start of the experiment were associated with a slightly reduced probability of CRISPR evolution by 3 dpi. Notably, at this final time point, measured final phage and cell densities had far larger impacts on CRISPR evolution than the initial phage inoculum, and CRISPR evolution was predicted at substantially lower phage and cell densities than at 2 dpi. Glucose concentration was not retained in any model, suggesting that its impact on cell density was indeed the main driver of the effects seen. Collectively, these results indicate that conditions of high phage exposure, be it due to high dose of phage or high carrying capacity, are associated with higher levels of CRISPR immunity.

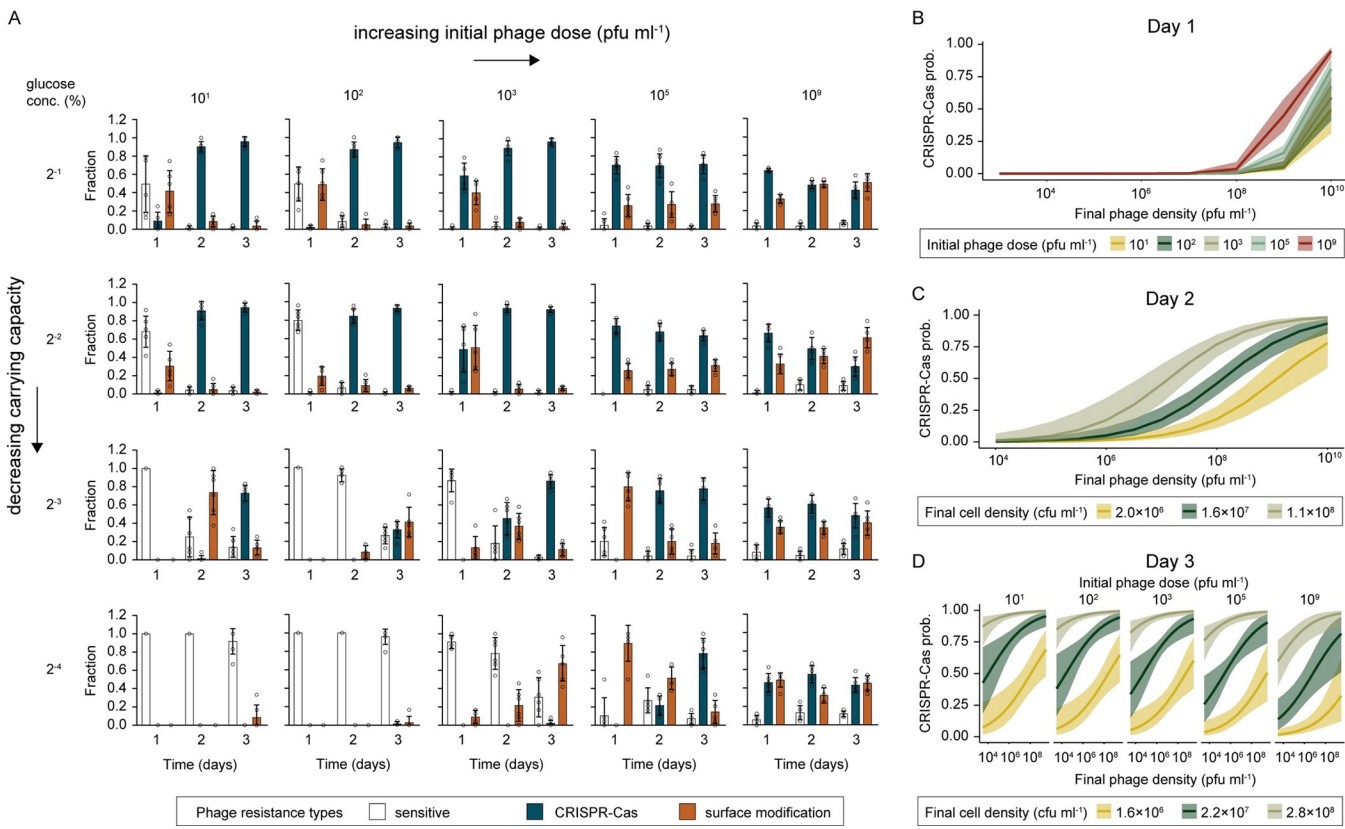

**Fig 3. Higher levels of CRISPR immunity are observed with higher phage exposure.** (**A**) Fraction of each resistance type (white: phage sensitive, blue: CRISPR-Cas immune, orange: surface-based resistance) over 3 days of evolution following exposure of initially phage sensitive WT *P. aeruginosa* to different amounts of DMS3*vir* phages ($10^1$, $10^3$, $10^5$, and $10^9$ PFU ml$^{-1}$) and in media containing different levels of glucose (0.2%, 0.02%, 0.002%, 0.0002% glucose, resulting in different carrying capacities; see S5A Fig). Data shown are the mean ± 1 standard deviation, 6 replicates per treatment, 24 clones tested per replicate. (**B–D**) Prediction plots showing model-estimated means and 95% confidence intervals based on statistical modelling of the data (in **A**), in which model selection was used to retain the most important predictors of CRISPR evolution on (**B**) Day 1, (**C**) Day 2, and (**D**) Day 3 (see S1 and S2 Tables). Data are available at https://doi.org/10.5281/zenodo.8193506.

## Discussion

CRISPR-Cas immune systems are abundant in nature, yet bacteria often evolve phage resistance through receptor mutation instead. Hence, it remains unclear when and where CRISPR-Cas systems play an important role in mediating phage defence [15]. The resistance type that dominates is predicted to be consequential for the community and may influence the long-term maintenance of CRISPR [16], since CRISPR clones pay an inducible cost and act as a phage sink, removing phages from the environment and allowing sensitive bacteria without resistance to invade. Moreover, whether bacteria evolve CRISPR-Cas or *sm* resistance against phages can have major implications for bacterial pathogenicity when in a host, since evolution of *sm* resistance (through the functional loss of the phage receptor that is also important for virulence) has been reported to cause virulence trade-offs that are not detected when the bacteria evolve CRISPR-based immunity [9].

Here, we combined theory and experimental work to examine the transient evolutionary dynamics of CRISPR-Cas (induced) and *sm* (constitutive) defences. We developed a model where we compete different bacterial genotypes that either carry or lack the resistance at each of 2 resistance loci, resulting in 4 distinct bacterial genotypes. This model allows us to track the transient dynamics of different resistance genotypes in combination with phage density. The

novelty of our approach is the population genetics perspective, which allows us to identify selection coefficients associated with each resistance mechanism. Our analysis shows how these selection coefficients vary over time due to the phage density dynamics driven by the proportion of resistant hosts in the bacterial population. In particular, our model shows how the emergence of the first resistance mechanism interferes with the evolution of the alternative resistance and may thus affect the long-term evolutionary outcome. This complements previous work that examined evolutionary stable strategies for investment in inducible and constitutive defences (i.e., long-term evolutionary outcomes), which showed that the frequency of infection has a major impact on the type of resistance that will dominate in the long term [4,7] and extends previous studies that modelled the short-term evolutionary dynamics of CRISPR immunity and surface-based resistance in bacterial populations containing phages [17,18], which showed that the amount of bacterial replication, or the mutation rate, was important for the levels of *sm* detected [18].

We used the *P. aeruginosa* and phage DMS3*vir* model system to test key predictions from our model in relation to the factors that drive the relative abundance of CRISPR immunity and *sm* resistance that emerge in phage-sensitive bacterial populations. This analysis showed the influence of phage exposure and the productivity of the environment on both the short-term and the long-term coevolutionary outcomes between these 2 alternative resistance strategies. We see that *sm* arises independent of phage infection but during DNA replication and we predicted, and experimentally demonstrated, that *sm* relative abundance in the population is dependent on the replication potential of the populations. Our finding that bacteria increasingly rely on their CRISPR-Cas immune systems under conditions of low bacterial growth is consistent with previous studies showing that CRISPR-Cas immune systems become relatively more important when the focal species is cultured in resource-limited growth media [4], when they are exposed to bacteriostatic antibiotics [19] or when they compete with other bacteria [9]. On the other hand, CRISPR immunity evolution is dependent on phage infection. Hence, increasing the cell culture carrying capacity and the number of phages initially present resulted in faster phage epidemics and, hence, greater phage exposure. The finding that higher phage densities promote evolution of CRISPR immunity is consistent with the positive correlation between CRISPR and phage prevalence in metagenome sequence data [20].

Even though high phage densities fuel the rate at which CRISPR immunity is acquired, bacteria that evolved *sm* resistance will dominate at very high phage densities, due to the compounding costs associated with CRISPR immunity that are induced by infection [4,10]. Indeed, our statistical model and experimental data support the notion that if phage titres are high following the emergence of resistance types, selection favours the invasion of *sm* resistance; for example, at the highest phage exposure treatments, bacterial cultures grown with 0.2% or 0.02% glucose contained approximately equal proportions of CRISPR immune bacteria and surface mutants, whereas cultures were dominated by CRISPR immune bacteria in the treatments with lower phage exposures.

While our model and experiments are useful for identifying ecological factors that shape the evolution of CRISPR-Cas and *sm* resistance when bacteria are exposed to a single type of phage, more complex models would be needed to consider scenarios where bacteria are exposed to genetically diverse phage populations, multiple phages and/or where phages can evolve in response to bacterial immunity. A previous coevolutionary model predicts that *sm* will be dominant in most conditions, due to negative selection imposed by phage escapers [18]. This is consistent with empirical studies showing that an increase in phage genetic diversity favours the evolution of *sm* resistance (which provides broad-range resistance) over CRISPR immunity (which is sequence specific) [21] and therefore suggests that larger phage

inoculums (which have greater standing genetic variation) might result in a dampened increase in CRISPR immunity evolution compared to our model predictions. Long-term coculture experiments with *P. aeruginosa* and phage DMS3*vir* show that evolution of CRISPR escaper phages is constrained by the natural evolution of high diversity in spacer repertoires of the bacteria [22,23]. This is consistent with models that predict that bacteria–phage coexistence and coevolution is highly sensitive to phage mutation rates and CRISPR immunity activity and acquisition rates [24–29]. Long-term ongoing CRISPR–phage coevolution has so far only been observed for *Streptococcus thermophilus* and its virulent phage 2972, resulting in an arms race dynamics that may ultimately lead to phage extinction as phage accumulate costly mutations and face an increasingly diverse spacer repertoire in the bacterial population [30–32]. A number of models have been developed to explore conditions where bacteria with CRISPR immunity and their phage can coexist, with or without coevolution [33–36], which may be mediated through CRISPR loss [16,37], exposure to a greater number of diverse phage species [38], or a spatial organisation of the bacteria and phage [28,39,40]. Future empirical studies are needed to explore patterns of CRISPR–phage coexistence and coevolution in environments with greater ecological complexity.

## Methods

### Mathematical model

See **S1 Text** and **S1–S4** Figs.

### Bacterial strains and phage

Bacterial strains used in this study include *P. aeruginosa* UCBPP-PA14 (WT), UCBPP-PA14 *csy3*::*lacZ* (KO) [14], UCBPP-PA14 BIM2 with 2 spacers targeting DMS3*vir* (BIM) [4], and UCBPP-PA14 *csy3*::*lacZ* spontaneous surface mutant (*sm*) [4]. Phages used in this study include the obligatory lytic temperate phage, DMS3*vir* [14], and DMS3*vir* carrying anti-CRISPR (Acr) IF1 [41].

### Experimental evolution

Evolution experiments with PA14 (WT) and DMS3*vir* were performed as previously described [4]. Briefly, 6 ml cultures of M9 containing 0.2%, 0.02%, 0.002%, or 0.0002% glucose in glass vials ($n = 6$) were inoculated with approximately $10^6$ colony-forming units (CFUs) of PA14 (1:1,000 subculture of M9 (0.2% glucose) adapted cells). Changing the glucose concentration changes the carrying capacity of the media, but growth rate is unaffected (**S5A Fig**). Phages were added to each vial in varying amounts ($10^1$, $10^2$, $10^3$, $10^5$, and $10^9$ PFUs), before the vial lids were tightly closed and cultures were incubated at 37 degrees with shaking. Cultures were subcultured 1:100 daily, for 3 days, and CFU and PFU counts were determined daily by plating and spot assays (**S5B** and S5C **Fig**). To determine the phage resistance phenotypes, 24 clones were randomly selected from each replicate, inoculated into LB in a 96-well plate and grown overnight. Cultures were streaked against phages DMS3*vir* and DMS3*vir*-AcrIF1. Consistent with previous work [4,42,43], phage-sensitive clones were susceptible to both phages, *sm* were resistant to both phages, and the CRISPR clones were resistant to DMS3*vir*, but not DMS3*vir*-AcrIF1. To test the effect of different inoculum amounts, varying amounts of 6 ml M9 (0.2% glucose) cultures were subcultured (600 μl: 10%, 60 μl: 1%, 6 μl: 0.1%, and 0.6 μl: 0.01%) into fresh media, for a total final volume of 6 ml. $10^5$ PFUs were added to each culture ($n = 6$), and phage resistance profiles were determined following 1 day of growth.

## Statistical modelling

Mixed effects models were constructed to examine the relative contributions of all potential predictors on CRISPR evolution. A binomial dataset was constructed where CRISPR evolution was coded as 1 or 0 for each clone per replicate. Next, a binomial generalised linear mixed effects model with fixed effects of glucose, cell density, phage density and initial phage inoculum, and treatment replicate as a random effect, was run for each time point (days 1 to 3). Cell and phage density data were log transformed. A maximal model was generated and all possible candidate models were compared using the AIC method with *dredge* from the MuMIn package [44]. AIC values assess the fit of a model by looking at the likelihood of a model given the data, penalising for increased number of parameters (as increased complexity of the model increases parameter uncertainty). We selected the most parsimonious model (i.e., the model with fewest parameters) within 2 delta AICs for each time point. Model comparisons based on AIC are presented in S1 Table, with the selected models highlighted. Model estimates for the selected models are presented in S2 Table.

For the selected models, prediction data frames with 95% confidence intervals were generated using *ggpredict* from the package ggeffects [45], model dispersion was tested and scaled residuals were examined using DHARMa residual diagnostics [46], and the final predictions were visualised with ggplot2 v3.3.2 and the wesanderson package. All statistical analyses were performed in R v4.0.2, and code and data are available at https://doi.org/10.5281/zenodo.8193506.

## Supporting information

**S1 Text. Description of the mathematical model.**
(DOCX)

**S1 Fig. Emergence of phage resistance over time.** Frequency of the different bacterial resistance types ($S$ (sensitive, white bars), $C$ (CRISPR immune, blue), $R$ (surface mutants, orange), or $D$ (both CRISPR and surface mutants, red)) through time (3 days) and for different values of the initial doses of free viruses ($V$) and the carrying capacity ($K$). All the simulations started with an initial density of susceptible cells at $K/100$. Other parameter values: $r = 1$, $m = 0$, $m_v = 0$, $a = 10^{-8}$, $B = 100$, $c_R = 0.01$, $\tau = 0.01$, $\mu = 10^{-4}$, $A = 5 \ 10^{-4}$, $L = 10^{-3}$.
(TIF)

**S2 Fig. Emergence of resistance over time in the absence CRISPR toxicity.** Frequency of the different bacterial resistance types ($S$ (sensitive, white bars), $C$ (CRISPR immune, blue), $R$ (surface mutants, orange), or $D$ (both CRISPR and surface mutants, red)) through time (3 days) and for different values of the initial doses of free viruses ($V$) and the carrying capacity ($K$), in the absence of an induced CRISPR immunity toxicity ($\tau = 0.0$). All the simulations started with an initial density of susceptible cells at $K/100$. Other parameter values: $r = 1$, $m = 0$, $m_v = 0$, $a = 10^{-8}$, $B = 100$, $c_R = 0.01$, $\tau = 0.0$, $\mu = 10^{-4}$, $A = 5 \ 10^{-4}$, $L = 10^{-3}$.
(TIF)

**S3 Fig. Effect of varying the initial densities of susceptible cells.** Plots show: (**A**, **D**) the predicted resistance type fractions, (**B**, **E**) cell counts, and (**C**, **F**) phage counts, with (**A-C**) the same initial phage dose ($V = 10^6$) or (**D-F**) when the initial dose of viruses was diluted to keep the same initial multiplicity of infection of 0.1. Amount of starting inoculum: 10% ($K/10$), 1% ($K/100$), 0.1% ($K/1,000$), 0.01% ($K/10,000$) of total final volume. Other parameter values: $r = 1$, $m = 0$, $m_v = 0$, $a = 10^{-8}$, $B = 100$, $c_R = 0.01$, $\tau = 0.01$, $\mu = 10^{-4}$, $A = 5 \ 10^{-4}$, $L = 10^{-3}$. Panels (**A-C**) are also shown in Fig 2, alongside the experimental results (Fig 2D–2F). Data are available at https://doi.org/10.5281/zenodo.8193506.
(TIF)

**S4 Fig. Transitory dynamics of bacteria and viruses across time.** The plots show the dynamics for 3 transfers (20 hours between each transfer, indicated by a vertical grey line), including (**A**) density of $S$ cells (susceptible, blue line), (**B**) density of $V$ (viruses, black line), (**C**) densities of the different types of resistant cells ($R$ (surface mutation, green line), $C$ (CRISPR immune, red line), and $D$ (both resistances, brown line), (**D**) frequencies of $R$, $C$, and $D$ cells (dashed lines with same colours as in panel C), and (**E**) dynamics of linkage disequilibrium between the resistance loci across time. Linkage disequilibrium between the resistance loci is measured as: $LD = f_S f_D - f_R f_C$. Parameter values: $r = 1$, $m = 0$, $m_v = 0$, $a = 10^{-8}$, $B = 100$, $c_R = 0.01$, $\tau = 0.01$, $\mu = 10^{-4}$, $A = 5\ 10^{-4}$, $L = 10^{-3}$, $K = 10^8$, and (initial) $V = 10^4$.
(TIF)

**S5 Fig. Cell and phage counts from evolution experiment with a range of phage doses and glucose treatments ([Fig 3]).** Plots show (**A**) growth curves of WT cultures grown in M9 containing different amounts of glucose (0.2%, 0.02%, 0.002%, and 0.0002%), (**B**) cell counts (CFU ml$^{-1}$), (**C**) phage counts (PFU ml$^{-1}$), and (**D**) the multiplicities of infection (MOI: phage count/cell count) across the 3-day experiment. Initial (day 0) values are depicted as white bars, increasing colour density (light-dark shades) represent increasing glucose concentrations in the growth media/culture carrying capacity. Data are available at https://doi.org/10.5281/zenodo.8193506.
(TIF)

**S6 Fig. Higher levels of CRISPR immunity are observed with higher phage exposure (selection of data from [Fig 3A]).** Fraction of each resistance type (white: phage sensitive, blue: CRISPR-Cas immune, orange: surface-based resistance) that evolved after 1 day of evolution following exposure of initially phage sensitive WT *P. aeruginosa* to (**A**) different amounts of DMS3*vir* phages ($10^1$, $10^3$, $10^5$, and $10^9$ PFU ml$^{-1}$) in media containing 0.2% glucose (highest carrying capacity) and (**B**) a moderate phage dose ($10^3$ PFU ml$^{-1}$) in media containing different levels of glucose (0.2%, 0.02%, 0.002%, 0.0002% glucose, resulting in different carrying capacities; see S4A Fig). Data shown are the mean ± 1 standard deviation, 6 replicates per treatment, 24 clones tested per replicate. Data are available at https://doi.org/10.5281/zenodo.8193506.
(TIF)

**S1 Table. AIC selection tables for binomial generalised linear mixed effects models.** Fixed effects of glucose, cell density, phage density and initial phage inoculum, and treatment replicate as a random effect, are shown for each time point. Rows with selected models are shown in bold. "Phage" refers to initial phage inoculum size, while "log_phage" and "log_cell" indicate measured phage and cell densities, respectively.
(DOCX)

**S2 Table. Estimates for binomial generalised linear mixed effects models with variables that were retained following AIC selection.** "Phage" refers to initial phage inoculum size, while "log_phage" and "log_cell" indicate measured phage and cell densities, respectively, at each time point.
(DOCX)

## Author Contributions

**Conceptualization:** Bridget Nora Janice Watson, Sylvain Gandon, Edze Rients Westra.

**Data curation:** Bridget Nora Janice Watson.

**Formal analysis:** Bridget Nora Janice Watson, Elizabeth Pursey.

**Funding acquisition:** Edze Rients Westra.

**Investigation:** Bridget Nora Janice Watson, Sylvain Gandon.

**Methodology:** Bridget Nora Janice Watson, Elizabeth Pursey, Sylvain Gandon.

**Supervision:** Edze Rients Westra.

**Visualization:** Bridget Nora Janice Watson, Elizabeth Pursey.

**Writing – original draft:** Bridget Nora Janice Watson, Sylvain Gandon, Edze Rients Westra.

**Writing – review & editing:** Bridget Nora Janice Watson, Elizabeth Pursey, Sylvain Gandon, Edze Rients Westra.

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
