## [Editor Report · Decision Letter 0]

12 Apr 2023

Dear Dr. Watson, 

Thank you for submitting your manuscript entitled "Eco-evolutionary feedbacks shape the evolution of constitutive and inducible defences" for consideration as a Research Article by PLOS Biology.

Your manuscript has now been evaluated by the PLOS Biology editorial staff, as well as by an academic editor with relevant expertise, and I am writing to let you know that we would like to send your submission out for external peer review.

Once your full submission is complete, your paper will undergo a series of checks in preparation for peer review. After your manuscript has passed the checks it will be sent out for review. To provide the metadata for your submission, please Login to Editorial Manager (https://www.editorialmanager.com/pbiology) within two working days, i.e. by Apr 14 2023 11:59PM.

Kind regards,

Paula

---

Senior Editor

PLOS Biology

---

## [Decision Letter · Decision Letter 1]

12 Jun 2023

Dear Dr. Watson,

Thank you for your patience while your manuscript "Eco-evolutionary feedbacks shape the evolution of constitutive and inducible defences" went through peer-review at PLOS Biology. Your manuscript has now been evaluated by the PLOS Biology editors, an Academic Editor with relevant expertise, and by several independent reviewers.

In light of the reviews, which you will find at the end of this email, we are pleased to offer you the opportunity to address the comments from the reviewers in a revision that we anticipate should not take you very long. Please also address the comments from the Academic Editor that you can find at the end of this letter. We will then assess your revised manuscript and your response to the reviewers' comments with our Academic Editor aiming to avoid further rounds of peer-review, although might need to consult with the reviewers, depending on the nature of the revisions.

Please also address the following policy and formatting requests:

1. DATA POLICY:

Regardless of the method selected, please ensure that you provide the individual numerical values that underlie the summary data displayed in the following figure panels as they are essential for readers to assess your analysis and to reproduce it: Figures 2ABCDEF, 3ABCD, Supplementary Figures S1, S2, S3ABCDEF, S4ABCDE, S5ABCD, S6AB.

**Please also ensure that figure legends in your manuscript include information on where the underlying data can be found, and ensure your supplemental data file/s has a legend.**

2. Please provide a blurb which (if accepted) will be included in our weekly and monthly Electronic Table of Contents, sent out to readers of PLOS Biology, and may be used to promote your article in social media. The blurb should be about 30-40 words long and is subject to editorial changes. It should, without exaggeration, entice people to read your manuscript. It should not be redundant with the title and should not contain acronyms or abbreviations.

3. We suggest a change in the title: "Transient eco-evolutionary dynamics early in a phage epidemic have strong and lasting impact on the long-term evolution of bacterial defenses"

**IMPORTANT - SUBMITTING YOUR REVISION**

*Resubmission Checklist*

*Published Peer Review*

*PLOS Data Policy*

*Blot and Gel Data Policy*

Sincerely,

Paula

---

Senior Editor

PLOS Biology

REVIEWS:

Reviewer #1: Evolution bacteria and phages.

Reviewer #2: Eco-evo of virus-host defence and mathematical modelling.

Reviewer #1: The interesting paper by Watson et al combines theoretical modeling with experimental evolution to study the intriguing problem of the choice bacteria make between two distinct immune strategies: innate immunity (surface receptor mutation) and adaptive immunity (CRISPR). The two mechanisms are under strong negative epistasis and so are effectively mutually exclusive. The theory and experiment converge at a rather intuitive solution: high level of bacterial reproduction favors receptor mutation because the mutation rate is proportional to the replication rate whereas low level of replication and high virus load in the initial stages of infection favor CRISPR defense.

This is an interesting and useful piece of work. Both the model and the experiment are quite straightforward, and the logic of the study is transparent, so I do not have major criticisms.

I would only note that I would avoid claiming that the theory developed here is "novel". The model as such may be original but the approach is quite simple and rather standard.

What is disappointing about the current manuscript is the discussion, which is basically 4limited to the explanation of the results of the present work rather than placing it in a more general context.

Accordingly, there are very few references. I believe a more general, conceptual discussion is highly desirable. Just as an example, it seems very strange that previous mathematical models of CRISPR evolution are not discussed:

https://pubmed.ncbi.nlm.nih.gov/23221803/

https://pubmed.ncbi.nlm.nih.gov/30905282/

https://pubmed.ncbi.nlm.nih.gov/34819498/

https://pubmed.ncbi.nlm.nih.gov/36645771/

Reviewer #2: The focus of this study is how the relative abundance of alternative defence systems is influenced by growth conditions. The model system is the bacterium Pseudomonas aeruginosa, its lytic virus DMS3vir, and two defence mechanism: A constitutive surface mutation and an inducible CRISPR-Cas system. A mathematical model is developed and used to discusscompared to the experimental results. A main finding, consistent with the model, is how the surface mutation requires cell divisions to become dominant while the CRISPR system requires phage infections. I find this an interesting and well-designed study adressing a sub-question of the general issue of how cost of resistance affect bacterial communities.

I appreciate the concise and relatively short introduction and discussion, but wonder if there are some points that deserve a bit more attention.

1 Constitutive versus inducible systems: Resistance acquired through the CRISPR-Cas system is of course inducible. But the cost of running the CRISPR system would seem to me to be constitutive as it seems to be linked to the production of the CAS proteins (Vale et al. https://doi.org/10.1098/rspb.2015.1270). I could not quite figure out whether this is relevant for this study and should be mentioned.

2. The relatively short discussion gives little room for a discussion of the consequences of the study. Cost of resistance and the associated trade-off between competition and defense is probably a main structuring effect of bacterial communites, both at the within-species (strain) level and the between-species (community composition level). If I got the actual model system, there would be a cost (fitness loss) from the surface mutation only in the mutants carrying this mutation. If the CRISPR system is expressed (produces CAS proteins) in all strains, the associated cost is at the species level, but the small additional cost of including the recognition sequence for DMS3vir would be at strain level. In the general context of host-virus interactions, I find this important and interesting, but I acknowledge the lack of space for complicated discussions.

COMMENTS FROM THE ACADEMIC EDITOR:

- I appreciate that the number of replications is important for resistance evolution vs CRISPR. However, it would be great if the authors discussed 1) the role of initial population size in the light of an initially higher pool of individuals that could obtain mutations, and 2) the role of the environment being "sparser" at lower glucose concentrations.

- The code should ideally be archived and receive a doi, e.g., via Zenodo.

(Potential) typos:

- l. 60 “allows for

- l. 78 “resistant”

- y axis Fig. 3 - “carrying capacity”?"

---

## [Editor Report · Decision Letter 2]

7 Aug 2023

Dear Dr Watson,

Thank you for the submission of your revised Research Article "Transient eco-evolutionary dynamics early in a phage epidemic have strong and lasting impact on the long-term evolution of bacterial defenses" for publication in PLOS Biology. On behalf of my colleagues and the Academic Editor, Claudia Bank, I am pleased to say that we can in principle accept your manuscript for publication, provided you address any remaining formatting and reporting issues. These will be detailed in an email you should receive within 2-3 business days from our colleagues in the journal operations team; no action is required from you until then. Please note that we will not be able to formally accept your manuscript and schedule it for publication until you have completed any requested changes.

PRESS

Sincerely,

Paula

---

Senior Editor

PLOS Biology
